# Intra- and Early Postoperative Evaluation of Malperfused Areas in an Irradiated Random Pattern Skin Flap Model Using Indocyanine Green Angiography and Near-Infrared Reflectance-Based Imaging and Infrared Thermography

**DOI:** 10.3390/jpm12020237

**Published:** 2022-02-08

**Authors:** Wibke Müller-Seubert, Patrick Ostermaier, Raymund E. Horch, Luitpold Distel, Benjamin Frey, Aijia Cai, Andreas Arkudas

**Affiliations:** 1Laboratory for Tissue Engineering and Regenerative Medicine, Department of Plastic and Hand Surgery, University Hospital Erlangen, Friedrich Alexander University Erlangen-Nuremberg FAU, 91054 Erlangen, Germany; patrickostermaier@yahoo.com (P.O.); raymund.horch@uk-erlangen.de (R.E.H.); aijia.cai@uk-erlangen.de (A.C.); andreas.arkudas@uk-erlangen.de (A.A.); 2Department of Radiation Oncology, University Hospital Erlangen, Friedrich Alexander University Erlangen-Nuremberg FAU, 91054 Erlangen, Germany; luitpold.distel@uk-erlangen.de; 3Translational Radiobiology, Department of Radiation Oncology, University Hospital Erlangen, Friedrich Alexander University Erlangen-Nuremberg FAU, 91054 Erlangen, Germany; benjamin.frey@uk-erlangen.de

**Keywords:** irradiation, imaging, malperfusion

## Abstract

Background: Assessment of tissue perfusion after irradiation of random pattern flaps still remains a challenge. Methods: Twenty-five rats received harvesting of bilateral random pattern fasciocutaneous flaps. Group 1 served as nonirradiated control group. The right flaps of the groups 2–5 were irradiated with 20 Gy postoperatively (group 2), 3 × 12 Gy postoperatively (group 3), 20 Gy preoperatively (group 4) and 3 × 12 Gy preoperatively (group 5). Imaging with infrared thermography, indocyanine green angiography and near-infrared reflectance-based imaging were performed to detect necrotic areas of the flaps. Results: Analysis of the percentage of the necrotic area of the irradiated flaps showed a statistically significant increase from day 1 to 14 only in group 5 (*p* < 0.05). Indocyanine green angiography showed no differences (*p* > 0.05) of the percentage of the nonperfused area between all days in group 1 and 3, but a decrease in group 2 in both the left and the right flaps. Infrared thermography and near-infrared reflectance-based imaging did not show evaluable differences. Conclusion: Indocyanine green angiography is more precise in prediction of necrotic areas in random pattern skin flaps when compared to hyperspectral imaging, thermography or clinical impression. Preoperative fractional irradiation with a lower individual dose but a higher total dose has a more negative impact on flap perfusion compared to higher single stage irradiation.

## 1. Introduction

Defect reconstruction still remains one of the main fields of interest in plastic surgery. Assessment of flap viability has been shown to be effective using the correct length-to-width ratio in the case of random pattern flaps for clinical evaluation [1,2].

When multimorbid oncological patients who might have undergone irradiation preoperatively or should undergo irradiation postoperatively, further reliable assessment of flap perfusion is necessary to avoid complications such as wound healing disorders and partial or complete flap loss [1]. Therefore, the correct intraoperative detection of malperfused areas is one of the points of interest [1,3]. Skin is especially one of the common areas for collateral injury caused by radiotherapy and increasingly in diagnostic radiology by the use of fast multislice CT scanners and fluoroscopically guided interventions [4]. Skin tolerance has often been the limiting factor in radiotherapy [5]. Currently, skin injuries lead to long-lasting burdens for the patients being treated.

Irradiation of tissue leads to ischemia and finally results in tissue fibrosis [6]. Briefly, irradiation induces DNA damage and the generation of reactive oxygen species leading to activation of the immune system and the recruitment of inflammatory cells. These neutrophils release additional inflammatory mediators. Furthermore, lymphocytes and monocytes migrate to the irradiated tissue. Monocytes differentiate into macrophages, which—together with other cells such as fibroblasts—release the transforming growth factor-beta (TGF-β), which stimulates fibroblasts to differentiate into myofibroblasts. In the end, myofibroblasts secrete extracellular matrix proteins and lead to increased tissue stiffness [7]. Besides these factors, vascular injury and changes in microvascular function are a primary pathogenetic signal for the procedure of fibrosis [8]. Early inflammatory vascular changes such as an increase in leukocyte adhesion due to irradiation can be seen within one hour after irradiation [9]. The developed tissue fibrosis itself reduces the perfusion of the irradiated tissue [7,10]. Furthermore, irradiation seems to have a negative impact on flap dimension [11].

To determine tissue perfusion, different imaging modalities have been tested, such as laser Doppler flowmetry for cutaneous circulation [12], infrared thermography [13], indocyanine green angiography [1,14] and near-infrared reflectance-based imaging [15]. While these devices may eventually predict the malperfusion accurately, there might be differences in their features such as application handling and invasivity.

This study evaluates the different imaging modalities—infrared thermography, indocyanine green angiography and near-infrared reflectance-based imaging [16,17]—to predict malperfused areas in an irradiated random pattern fasciocutaneous flap model.

## 2. Materials and Methods

Twenty-five male Lewis rats (age 14 ± 2.5 weeks (range 10–19 weeks)) weighing 336 ± 22.4 g (range 283–390 g) were operated in 5 different treatment groups. The study was approved by the ethic committee of the government of Middle Franconia (RUF-55.2.2-2532-2-1275-15).

### 2.1. Surgical Procedure

Anesthesia was performed using isofluran. For analgesia, the animals received butorphanol (0.5–2 mg per kg) and meloxicam (1 mg per kg). Two modified caudally based McFarlane flaps were harvested at the rat’s back with a length of 6 cm and a width of 1 cm. The flaps were located parallel and 1 cm lateral to the spine (Figure 1). The flaps were harvested by incision along their medial, lateral and cranial side so that their caudal base was 1 cm cranial of the spina iliaca posterior superior. The dissection was deep to the panniculus carnosus and superficial to the deep fascia. After raising the flap, it was reinserted to its bed and sutured using monofilament sutures. Postoperative analgesia was performed using meloxicam. The rats received an antibiotic treatment with enrofloxacin (7.5 mg per kg) for 5 days.

### 2.2. Irradiation Procedure

An orthovoltage X-ray device performed the irradiation with a current of 20 mA and a voltage of 150 kV. For the irradiation, the rats received intramuscular anesthesia using ketamine (100 mg per kg) and medetomidine (0.2 mg per kg). Rats were positioned on the belly and transferred to the irradiation unit in a closed isolation cage to protect the rats from pathogens. An area of 7 × 2 cm² including the right flap or the area of the prospective right flap in case of preoperative irradiation was irradiated (Figure 2). The rest of the body of the rat was covered with lead shields as protection from irradiation. The left flap served as nonirradiated internal control. Postoperative irradiation was performed on the first day after the operation. Postoperative fractional irradiation was performed on the first, second and third day after the operation. Single stage preoperative irradiation was performed 4 weeks before the operation and fractional preoperative irradiation on the three following days starting 4 weeks preoperatively (Figure 3).

### 2.3. Groups

Animals were divided in 5 groups (*n* = 5). Group 1 included the control group without irradiation. The rats of group 2 received postoperative irradiation with 20 Gy. The rats of group 3 had postoperative irradiation with 3 × 12 Gy. Rats of group 4 received preoperative irradiation with 20 Gy and rats of group 5 fractional preoperative irradiation with 3 × 12 Gy (Table 1).

### 2.4. Imaging

Indocyanine green angiography using an IC-Flow™ Imaging System (Diagnostic Green, Farmington Hills, MI, USA) was performed directly after flap harvest and on day 1, 7 and 14 after the operation in groups 1, 2 and 3 (Figure 4). Therefore, indocyanine green was injected via the tail vein (1 mg per kg). An intensity of less than 20% of the maximum intensity was defined as malperfused area.

Standard clinical imaging was performed directly after the operation and on day 1, 3, 7, 10 and 14 after the operation. The animals were sacrificed 14 days after the operation. As previously described [16], near-infrared reflectance-based imaging was performed on the same days using Snapshot NIR^®^ (Kent Imaging Inc.; Calgary, AB, Canada) to measure tissue oxygenation in superficial tissue. Near-infrared light is transmitted onto the skin surface and reflected off the blood within the tissue [16]. There is a difference of oxygenated and deoxygenated light absorption of hemoglobin that depends on the wavelength. In conclusion, the ratio from oxygenated to deoxygenated blood and therefore the viability can be determined by this method. Poorly perfused skin has a lower percentage of oxygenated hemoglobin than well-perfused skin [16,18]. Infrared thermography images were obtained on the same days by a smartphone-compatible thermographic camera (FLIR ONE Pro, FLIR Systems, Inc.; Wilsonville, OR, USA). The camera uses a long-wave infrared sensor. It has an effective temperature range from −20 to 400 °C with a resolution of 0.1 °C with a sensitivity that detects temperature differences as low as 70 mK. Image processing is used to merge the photo with a thermal image and thus measure the temperature [16].

### 2.5. Blood Samples

Blood samples were taken 30 min after irradiation in the postoperative irradiation groups for staining of DNA double-strand breaks in lymphocytes as previously described [19]. Briefly, lymphocytes were isolated using Pancoll Separating Solution (Pan Biotech, Aidenbach, Germany). After separation, the lymphocytes were incubated overnight with the antibody detecting the phosphorylated variant of the histone H2AX (γ-H2AX) (Purified anti-H2A.X Phospho (Ser139); BioLegend, San Diego, CA, USA).

### 2.6. Statistical Analysis

Flap size and malperfused/necrotic areas were calculated using Image J (U.S. National Institutes of Health, Bethesda, MD, USA). The statistical analysis was performed using Microsoft Excel (Microsoft, Redmond, WA, USA) and Prism 8 (GraphPad Software, San Diego, CA, USA). The normal distribution was identified graphically using QQ plots. For the comparison of the medians of the same group at different timepoints and the comparison of different groups at the same timepoint, mixed effect models with the Geisser–Greenhouse correction following the Tukey test were used. The level for statistical significance was set at *p* < 0.05.

## 3. Results

All animals tolerated the operative procedure without complications. Postoperative irradiation resulted in weight loss of approximately 10% (group 2, weight day at 0: 322 ± 20.3 g, weight at day 14: 297 ± 17.2 g; group 3 weight at day 0: 329 ± 17.8 g, weight at day 14: 296 ± 14.2 g). The weight of the rats in groups 1, 4 and 5 remained stable.

### 3.1. Flap Size

Mean flap size (including all well- and mal- or nonperfused areas) was slightly reduced (*p* > 0.05) 14 days after the operation in all groups (Figure 5).

### 3.2. Necrotic Areas of Flaps

There was no difference (*p* > 0.05) in the percentage of the necrotic area of the flaps between days 1 and 14 in the control group (comparison day 1 to 7: *p* = 0.04; day 1 to 10: *p* = 0.01). Furthermore, there was no difference between the control group and the treatment groups or between the right and left flaps within each group.

Comparison of the percentage of the necrotic area of the irradiated flaps showed an increase in all groups from day 1 to 14 (Figure 6). The difference between day 1 and 14 was statistically significant only in group 5 (comparison day 1 to 7: *p* = 0.03; day 1 to day 10: *p* = 0.01; day 1 to 14: *p* = 0.02).

### 3.3. Nonperfused Areas of the Flaps in the Indocyanine Green Angiography

The indocyaningreen angiography during the operation showed no differences (*p* > 0.05) of the mean percentage of the nonperfused area between all groups or between the left and the right flaps within each group (Figure 7). On days 1, 7 and 14, no difference (*p* > 0.05) of the percentage of the nonperfused area of the flaps in group 1 and 3 was measured. The percentage of the nonperfused area of the flaps in group 2 decreased between day 1 and 7 (*p* < 0.01) and between day 1 and 14 (*p* = 0.01). However, as the percentage of the nonperfused area in the intraindividual control group in group 2 decreased as well, there was no difference between the left and the right flap of group 2 on the days 1, 7 and 14.

### 3.4. Staining of Double-Strand Breaks

Staining of the lymphocytes, which were extracted from the blood samples taken 30 min after the end of the irradiation procedure, showed double-strand breaks in all groups with a lower number in group 2 and a higher number in group 3 (Figure 8). These double-strand breaks show that all irradiation regimens had an influence on the rat’s organism and that the irradiation was not just applied locally on the flaps.

### 3.5. Near-Infrared Reflectance-Based Imaging and Infrared Thermography

Near-infrared reflectance-based imaging did not produce valuable data in our study. As seen in Figure 9, the sutures and the wound scab at the sides did not allow for capturing the oxygen levels in these areas, thus making precise evaluation impossible.

Evaluation of the images of the infrared thermography did not show evaluable differences of the temperature. Due to the previous shaving and the hair loss after irradiation, the measured temperature was generally high without specific differences between the flaps, the surrounding tissue or between the different areas of the flap itself (Figure 10).

## 4. Discussion

In times of multimorbid patients undergoing complicated reconstructive procedures including reconstructive flap surgery [20], predictable complications such as partial flap necrosis due to malperfusion should be avoided. Replacing irradiated tissue with unirradiated flaps is one way to circumvent such surgical site sequelae [21,22,23]. To assess the viability and perfusion of flaps properly, different imaging modalities have been evaluated in this study in the setting of pre- and postoperative irradiation of random pattern skin flaps.

From the three imaging methods tested, the indocyanine green angiography was shown to be the most precise. The advantages of the infrared thermography such as its noninvasive use have been already shown in other studies [13]. In contrast to the study of Li et al.; infrared thermography was shown not to reproduce evaluable results in our study. One reason might be that the areas of interest—the flaps—were hairless during the total period of observation. Repetitive shaving was not possible due to the developed desquamation and necrosis. Hairless areas might be a necessary condition for infrared thermography [24].

While Chin et al. stated that prediction of region-of-flap necrosis might be possible due to the detection of early changes in deoxygenated hemoglobin seen by near-infrared reflectance-based imaging [15], we could not produce evaluable data. The reason for this might be similar to those of the limitations of the infrared thermography.

Indocyanine green angiography has been proven to predict flap survival area on the first postoperative day more accurately than other imaging modalities such as laser Doppler [1]. Especially, the early detection of malperfused areas of flaps is important to adapt the flap design at best intraoperatively, for example, by resection of the malperfused parts to prevent partial flap loss. Indocyanine green angiography supports the intraoperative decision-making process of flap design in daily clinical practice [3]. Giunta et al. stated that a perfusion index of less than 25% of the reference skin could be considered as a sign of developing flap necrosis [14], which is similar to the 20% we chose for our study. While the percentage of the necrotic area of the flaps increased in all groups, when evaluated clinically, the percentage of the nonperfused areas of the flaps—seen in the indocyanine green angiography—remained at least constant. These data support the assumption that indocyanine green angiography reliably foresees the well- and nonperfused areas of random pattern flaps. Interestingly, we measured a reduction of the nonperfused areas in group 2 after 20 Gy irradiation on days 7 and 14 compared to day 1. One reason might be that single irradiation with 20 Gy does not lead to harmful effects on flap perfusion. The improved perfusion might be due to the mechanism of flap delay, where linking or so-called choke vessels dilatate and reorientate due to increased blood flow because of opening of arteriovenous anastomoses and result in an enlarged perfused area [25,26]. As we measured the decrease in both the irradiated and the nonirradiated flaps without difference between these two groups, it might be possible that the animals in these groups were more susceptible to the changes of the delay phenomenon.

When comparing clinical evaluation of the necrotic area of the flaps to evaluation by indocyanine green angiography, the percentage generally turns out to be higher when compared to indocyanine green angiography. One reason for this difference might be that the indocyanine green angiography is able to evaluate the flap in its total thickness. In contrast, the clinical impression just assesses the visible superficial layers of the flap. It might be that the necrotic tissue, which was seen, does not affect all layers of the flap and is just necrotic scab as a sign of partial flap necrosis.

In addition to the imaging used in this study, duplex sonography may provide additional evidence of changes in flap perfusion. For example, Heitland et al. measured an increase in blood flow after anastomosis in musculocutaneous and perforator flaps compared to the donor vessel that indicates flap hyperperfusion [27]. Duplex sonography was not indicated in this model due to the random pattern perfusion. The irradiation had a systemic impact on the rats. This could be seen as DNA double-strand breaks were visible in all groups as proof of the systemic influence. Previous tests in our laboratory have shown that postoperative irradiation with 30 Gy was the lethal dose in our setting with a weight loss of more than 20% over several weeks. A reason might be that shorter, more intense irradiation treatments can increase acute effects because they do not allow sufficient time for regeneration [28].

Irradiation of the flaps resulted in a shrinking of the flaps in all groups, but the difference was not statistically significant. This might be explained by the fact that irradiation reduces the elasticity of the affected skin [5]. It is well known that tissue fibrosis is one of the hallmarks of chronic damage due to irradiation and changes the form, function and the appearance of the irradiated tissue [7]. In general, tissue fibrosis is a dynamic process with constant remodeling and long-term fibroblast activation. Initiation of these processes, however, could be seen 24 h after irradiation [29]. Especially early microvascular changes are seen within hours after irradiation [9]. As microvascular depletion leads to tissue ischemia, tissue hypoxia might be relevant for early fibrogenesis [8]. Skin reactions on irradiation such as desquamation were first visible at day 7 after irradiation with a peak at day 14 in a mouse model after 50 Gy irradiation [30]. The clinical sign of alopecia [5] or reduced hair growth due to irrradiation was seen in all flaps. High dose irradiation has been shown to cause dermal thickening [5], which unfortunately was not measured in our study. The shrinkage of the flap, however, is seen as sign of the developed fibrosis. Lin et al. performed fractioned irradiation with fractions up to 400 cGy and in total up to 40 Gy four weeks postoperatively [11]. The animals were killed six weeks after the end of irradiation. Greater tissue damage in both skin and muscle tissue of rat rectus muscle flaps with histological changes such as fibrosis and vascular damages were seen after irradiation with greater dose per fraction and less total dose compared to lower dose fraction groups and higher total dose. In our study, we did not see a statistically significant difference in all groups.

We did not see statistically significant differences in the size of the necrotic area between the treatment groups and the control group. However, we did see an increase of the necrotic area in all irradiated groups with a statistically significant increase in group 5. This could indicate that the damage of irradiation on tissue perfusion was not completely detected in our study setting. Similarly, postoperative irradiation of free skin flaps in rats one week after transplantation did not show statistical significant differences in the mechanical strength of the healing interface or the biochemical markers compared to the control group [31]. The irradiated flaps just showed minimal histomorphological changes compared to the nonirradiated flaps. Additionally, the irradiation with 20 Gy did not affect flap survival.

Furthermore, our study examines the acute phase after irradiation. It might be possible that the late harmful effects of irradiation are not included. As the study of Sumi et al. suggested, however, postoperative irradiation of skin flaps in rats that begins 3–4 weeks postoperatively should not affect flap survival negatively [32]. In this study, the postoperative irradiation begins at the first day after the operation. Therefore, one might expect that this study captures the majority of the negative side effects of irradiation.

In contrast to the harmful influence of irradiation observed in our study, low dose irradiation of ischemic pedicled skin flaps in mice with 5 Gy upregulates angiogenic chemokines and resulted in an increase of the vascularity of the flaps measured by laser Doppler. In addition, the irradiated flaps did not show an increase in flap ischemia or in flap necrosis compared to controls. Thus, it is suggested that neovascularization after ischemic injury is a multifactorial process [33]. Furthermore, low dose irradiation of ischemic legs in mice with 2 Gy promoted neovascularization by release of vascular endothelial growth factor (VEGF) [34]. Besides neovascularization, low dose fractionated irradiation with four fractions of 0.3 Gy in murine ischemic limbs improved capillary density and stimulated collateral vessel formation [35]. In conclusion, it can be assumed that the harmful effects of irradiation are directly related to the applied dose. For example, human endothelial cells have been shown to be viable up to an irradiation dose of 10 Gy [36]. However, it has been shown that irradiation with 40 Gy one month before harvesting an axial fasciocutaneous flaps resulted in an increased flap necrosis and a decrease in vascular density compared to the nonirradiated control group [37].

Limitations of this study include the small number of animals per group, which might be a reason for the results that showed just a few statistically significant differences. The irradiation regimen that was chosen in this study can only be transferred to clinical daily routine to a limited extent, since, for example, postoperative irradiation is usually only carried out after wound healing has been completed and not, as in this study, on the first postoperative day. The timing in this study was chosen so that the possible influence of the delay phenomenon could be kept as low as possible. Histopathological examinations or examinations on the molecular level, which could provide an indication of a possible neoangiogenesis, were not performed. This study deliberately wanted to focus on the use and comparison of imaging as it is also practiced in routine clinical practice. To distinguish necrotic areas in all layers from superficial necrosis more precisely, histopathological staining could be performed in future studies. In addition, the influence of irradiation could then also be studied at the molecular level. There is a wide range of different diagnostic procedures to measure tissue perfusion. For example, the measurement of transcutaneous oxygen pressure has been proven useful for the evaluation of free flap viability [38]. Skin perfusion pressure seems to be an accurate predictor of wound healing potential, especially in patients with limb ischemia [39]. In this study, we focused on three different imaging modalities that assessed the entire flap. Future studies could compare the indocyaningreen angiography, which in this study has been shown to be more precise, with other procedures.

## 5. Conclusions

This study shows that indocyanine green angiography is more precise in the prediction of necrotic areas in random pattern skin flaps when compared to hyperspectral imaging, thermography or clinical impression, since infrared thermography and near-infrared reflectance-based imaging did not produce reliable data in this setting. The impact of pre- and postoperative irradiation on flap perfusion seems to be the same. Furthermore, preoperative fractional irradiation with a lower individual dose but a higher total dose has a more negative impact on flap perfusion compared to higher single stage irradiation.

## Figures and Tables

**Figure 1 jpm-12-00237-f001:**
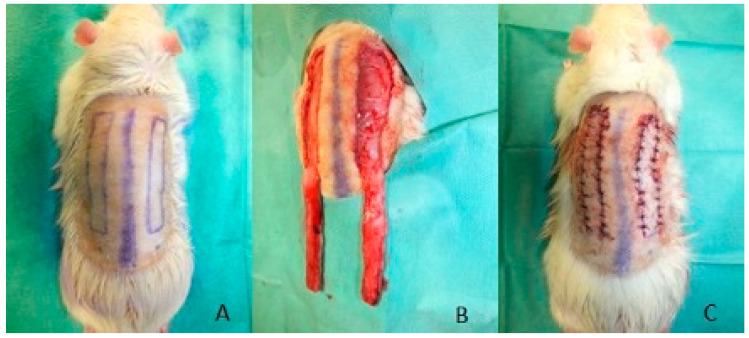
Harvest of two modified McFarlane flaps: (**A**) preoperative markings parallel and 1 cm lateral to the spine; (**B**) two flaps harvested; (**C**) reinsertion of the flaps.

**Figure 2 jpm-12-00237-f002:**
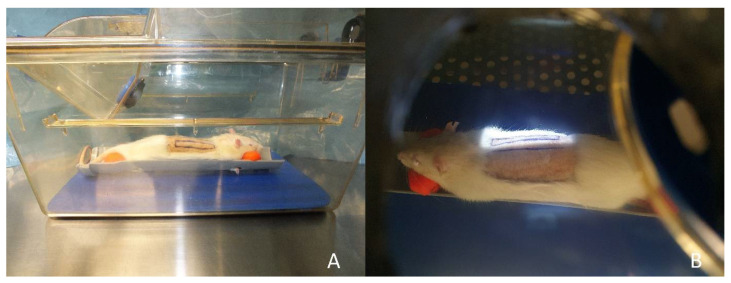
Irradiation procedure: (**A**) rat positioned for irradiation; (**B**) irradiated area.

**Figure 3 jpm-12-00237-f003:**
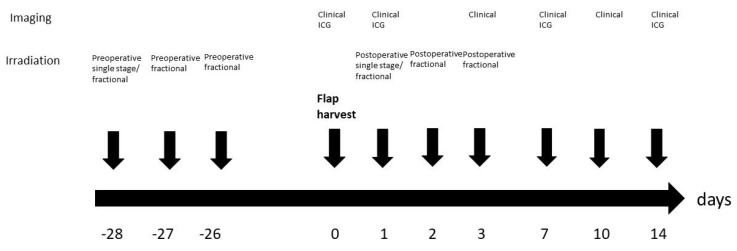
Study protocol.

**Figure 4 jpm-12-00237-f004:**
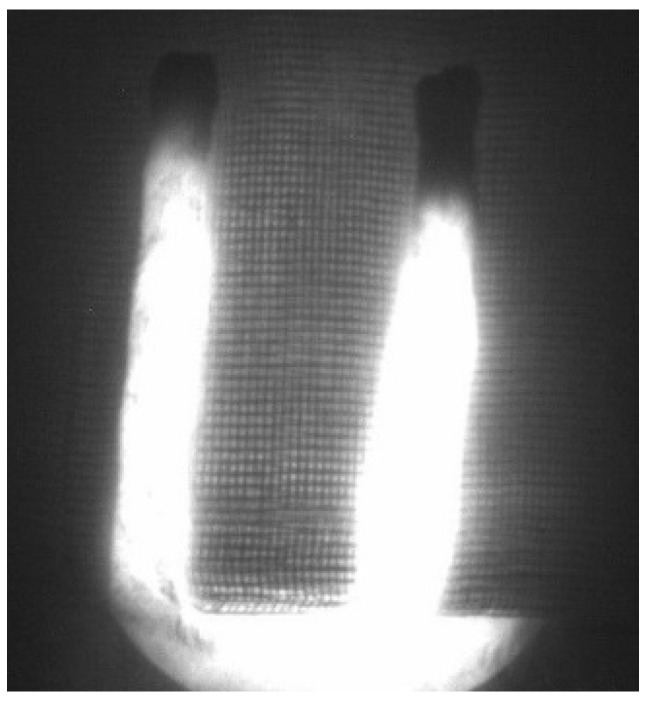
Indocyanine green angiography after flap harvest in group 2.

**Figure 5 jpm-12-00237-f005:**
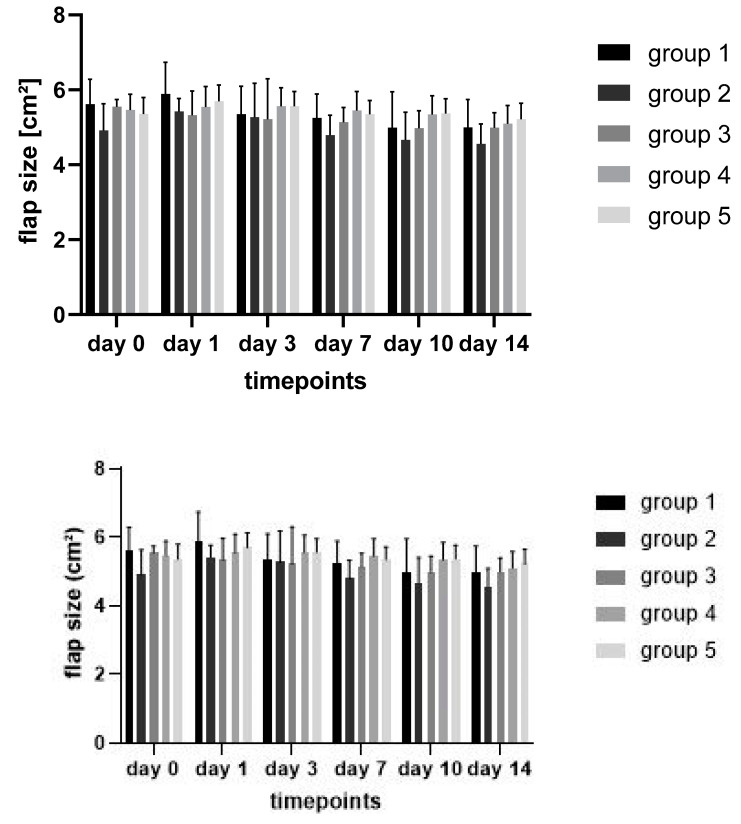
Flap size over time in the different treatment groups.

**Figure 6 jpm-12-00237-f006:**
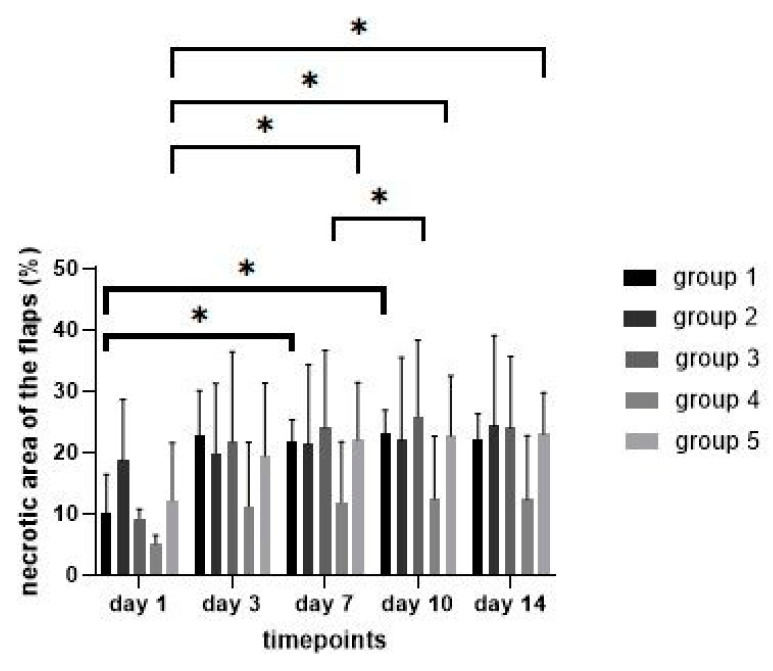
Necrotic area of the flaps over time in the different treatment groups. * showing statiscally significant differences (*p* ≤ 0.05).

**Figure 7 jpm-12-00237-f007:**
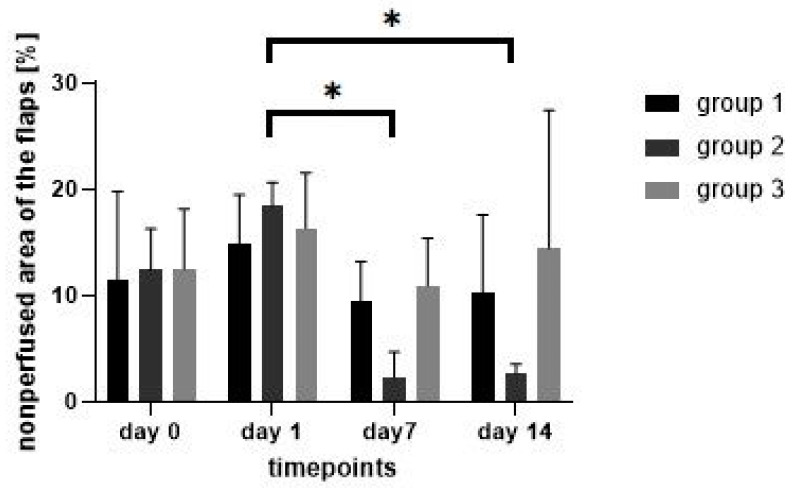
Nonperfused areas of the flaps in the indocyanine green angiography over time in the different treatment groups. * showing statiscally significant differences (*p* ≤ 0.05).

**Figure 8 jpm-12-00237-f008:**
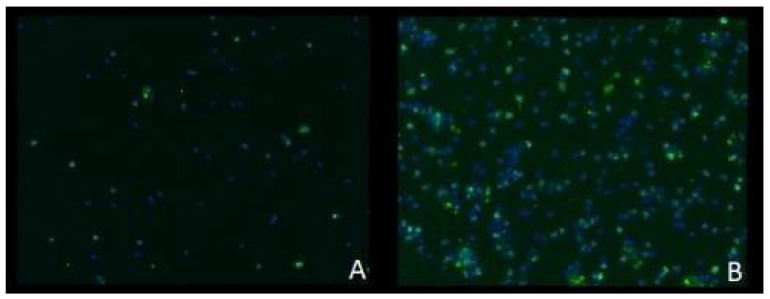
Staining of DNA double-strand breaks (green) in lymphocytes (blue) showing an increase in double-strand breaks after higher total dose irradiation: (**A**) group 2; (**B**) group 3.

**Figure 9 jpm-12-00237-f009:**
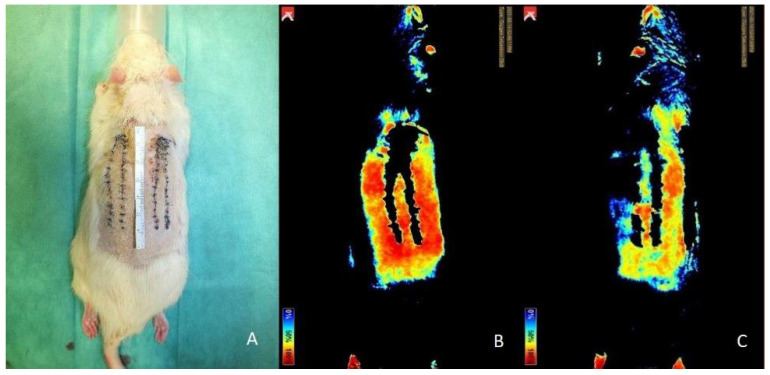
Group 2 comparison of clinical imaging (**A**) and near-infrared reflectance-based imaging at day 14: (**B**) right flap; (**C**): left flap.

**Figure 10 jpm-12-00237-f010:**
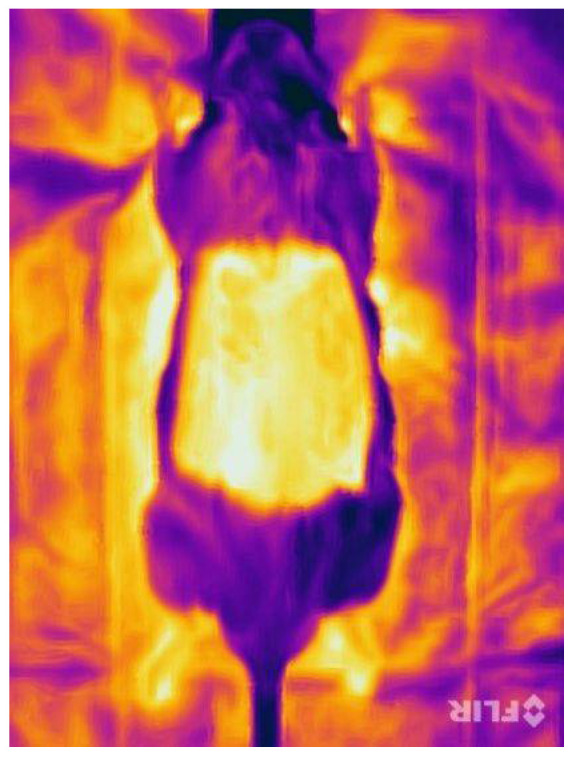
Infrared thermography on day 14 in group 2.

**Table 1 jpm-12-00237-t001:** Groups.

Group.	Irradiation Regimen
Group 1	None=control
Group 2	Single stage postoperative irradiation 20 Gy 1 day after flap harvest
Group 3	Fractional postoperative irradiation 12 Gy each day 1, 2 and 3 after flap harvest
Group 4	Single stage preoperative irradiation 20 Gy 28 days before flap harvest
Group 5	Fractional preoperative irradiation 12 Gy each day 28, 27 and 26 before flap harvest

## Data Availability

Data sharing is not applicable to this article.

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
