# Peer review of "Intra- and Early Postoperative Evaluation of Malperfused Areas in an Irradiated Random Pattern Skin Flap Model Using Indocyanine Green Angiography and Near-Infrared Reflectance-Based Imaging and Infrared Thermography"

_jpm, 2022, doi:10.3390/jpm12020237_

Round 1
Reviewer 1 Report
Thank you for let me reading this interesting paper where the authors explore different technology for evaluating the random pattern flap perfusion before after RT.
The study has been well projected and material and method are clear and described in details.
Results are well summarised and the discussion supports their results.
I would comment in the discussion that the experimental condition is far from the clinical practice especially because post op radiotherapy is normally postponed up the surgical wound is completely healed.
On the other had the article explore an important field in oncologic surgery and definitely helps to clarify the interaction between radiotherapy and random pattern flaps as well as provide evidence on the sensitivity of the different equipment available to investigate tissue perfusion.
Author Response
Point 1: I would comment in the discussion that the experimental condition is far from the clinical practice especially because post op radiotherapy is normally postponed up the surgical wound is completely healed.
Answer 1: You are absolutely right that in everyday clinical practice, irradiation is usually applied only after wound healing is complete. In this study we deliberately did not wait for this to happen in order to minimise the influence of the delay effect and possibly flap autonomisation. We have discussed this point (see red typed lines 364-372).
Reviewer 2 Report
Article: JPM-1491420
"Intra- and early postoperative evaluation of malperfused areas in an irradiated random pattern skin flap model using indocyanine green angiography and near-infrared reflectance based imaging and infrared thermography"
Comments to the Author
Many thanks for asking me to review the paper entitled ‘Intra- and early postoperative evaluation of malperfused areas in an irradiated random pattern skin flap model using indocyanine green angiography and near-infrared reflectance-based imaging and infrared thermography.’ The authors have attempted to evaluate the different imaging modalities infrared thermography, indocyanine green angiography, and near-infrared reflectance-based imaging to predict malperfused areas in an irradiated random pattern fasciocutaneous flap model, which is a crucial issue in the reconstructive surgery field. The study includes 25 McFarlane Flap models consisting of 5 different treatment groups. They conclude that indocyanine green angiography is more precise in predicting necrotic areas in random pattern skin flaps when compared to hyperspectral imaging, thermography, or clinical impression. In addition, they concluded that preoperative fractional irradiation with a lower individual dose but a higher total dose has a more negative impact on flap perfusion than higher single-stage irradiation. Overall, the manuscript is approaching an interesting topic in microsurgery and presents a timely research study. It allows researchers to set strategic directions for future research and provide practitioners with helpful guidance.
Major concerns:
I have a few comments and questions to help you improve your paper.
- As the authors commented, angiogenesis is fundamental to flap survival. The cell transduction signals for the stimulation of angiogenesis, including various growth factors, have recently aroused considerable interest. However, there has been no investigation on the growth factors or cytokines in the present study, including VEGF, bFGF, nor histologic assessment measuring the microvascular density from the experimental model. ELISAs or immunocytochemistry from the dorsal flap site of the specimen would have strengthened the authors' hypothesis. In addition, there exists a possibility that immune-mediated inflammation may be influencing or confounding the results; thus, it would be better to conduct immunohistochemical staining with the CD34, CD31, or CD14 endothelial markers. The authors would strengthen the quality of the paper by discussing the rationale for the selection of these methodologies.
- It would be better for the readers to disclose the brief mechanism of 'near-infrared reflectance-based imaging' or 'infrared thermography' to assess the flap circulation in this comparison study. It would also be curious to compare the transcutaneous oxygen pressure (TcPO2), skin perfusion pressure (SPP), or duplex sonography to the ICG, though not the aim of the paper, because those systems have been worked as the most reliable methods to predict the healing potential or flap survival.
Minor concerns:
- Some content in the Introduction (e.g., Lines 87-90) is more appropriate for Discussion.
- Minor typos: indoycyaningreen in Line 85

Author Response
Point 1: As the authors commented, angiogenesis is fundamental to flap survival. The cell transduction signals for the stimulation of angiogenesis, including various growth factors, have recently aroused considerable interest. However, there has been no investigation on the growth factors or cytokines in the present study, including VEGF, bFGF, nor histologic assessment measuring the microvascular density from the experimental model. ELISAs or immunocytochemistry from the dorsal flap site of the specimen would have strengthened the authors' hypothesis. In addition, there exists a possibility that immune-mediated inflammation may be influencing or confounding the results; thus, it would be better to conduct immunohistochemical staining with the CD34, CD31, or CD14 endothelial markers. The authors would strengthen the quality of the paper by discussing the rationale for the selection of these methodologies.
Answer 1: You are right that an additional examination for the expression of growth factors or immunohistochemical stainings with the question of, for example, angiogenesis would provide even more information. In this study, we first wanted to compare the different imaging methods that are currently predominantly used in clinical practice. An additional histopathological staining or an examination at the molecular level would be a good approach for a further study. We have taken up this point (see red typed lines 369-372).
Point 2: It would be better for the readers to disclose the brief mechanism of 'near-infrared reflectance-based imaging' or 'infrared thermography' to assess the flap circulation in this comparison study.
Answer 2: We have described the methods mentioned in the material section (see red typed lines 162-167, 169-172).
Point 3: It would also be curious to compare the transcutaneous oxygen pressure (TcPO2), skin perfusion pressure (SPP), or duplex sonography to the ICG, though not the aim of the paper, because those systems have been worked as the most reliable methods to predict the healing potential or flap survival.
Answer 3:Unfortunately, we could not compare all imaging modalities that currently exist in this study. We chose the three described because thermal imaging and near infrared reflectance based imgaing in particular seem promising as comparatively new and easy to perform methods. But it is indeed possible to test other imaging modalities in follow-up studies. We have mentioned this point as a possible field of interest for further studies (see red typed lines 375-381).
Point 4: Some content in the Introduction (e.g., Lines 87-90) is more appropriate for Discussion.
Answer 4: We modified the lines and have added them in the discussion (see red typed lines 309-311).
Point 5: Minor typos: indoycyaningreen in Line 85
Answer 5: Thank you very much, we corrected this spelling mistake (line 82).